# Hypovitaminosis D and Low T3 Syndrome: A Link for Therapeutic Challenges in Patients with Acute Myocardial Infarction

**DOI:** 10.3390/jcm10225267

**Published:** 2021-11-12

**Authors:** Alessandro Pingitore, Francesca Mastorci, Sergio Berti, Laura Sabatino, Cataldo Palmieri, Giorgio Iervasi, Cristina Vassalle

**Affiliations:** 1Clinical Physiology Institute, CNR, 56124 Pisa, Italy; mastorcif@ifc.cnr.it (F.M.); laura.sabatino@ifc.cnr.it (L.S.); iervasi@ifc.cnr.it (G.I.); 2Fondazione CNR-Regione Toscana G. Monasterio, 54100 Massa, Italy; berti@ftgm.it (S.B.); palmieri@ftgm.it (C.P.); cristina.vassalle@ftgm.it (C.V.)

**Keywords:** acute myocardial infarction, low T3 syndrome, hypovitaminosis D

## Abstract

Background and Aims: Vitamin D counteracts the reduction in the peripheral conversion of tiroxine (T4) into triiodothyronine (T3), which is the mechanism of low T3 syndrome (LT3) in acute myocardial infarction (AMI). The aim of this study was to assess the relationship between LT3 and hypovitaminosis D in AMI patients. Methods and Results: One hundred and twenty-four AMI patients were enrolled. Blood samples were taken at admission, and at 3, 12, 24, 48, and 72 h after admission. LT3 was defined as a value of fT3 ≤ 2.2 pg/mL, occurring within 3 days of hospital admission. Levels were defined as follows: sufficiency as a value of ±30 ng/mL, vitamin D insufficiency as 25-hydroxyvitamin D (25(OH)D) between 21 and 29 ng/mL, deficiency in 25(OH)D as below 20 ng/mL, and severe deficiency as values under 10 ng/mL. The percentage of subjects with severe 25(OH)D deficiency was significantly higher in the LT3 group (33% vs. 13%, *p* < 0.01). When LT3S was evaluated as a dependent variable, severe 25(OH)D deficiency (OR 2.6: 95%CI 1–6.7, *p* < 0.05) remained as an independent determinant after logistic multivariate adjustment together with age (>69 yrs, 50th percentile; OR 3.4, 95% CI 1.3–8.3, *p* < 0.01), but not female gender (OR 1.7, 95% CI 0.7–4.2, *p* = ns). Conclusions: This pilot study shows a relationship between hypovitaminosis D and LT3 in AMI patients. This association opens potential therapeutic challenges concerning the restoration of euthyroidism through vitamin D administration, together with the normalization of hypovitaminosis.

## 1. Introduction

Low triiodothyronine syndrome (LT3: reduced free T3 < 2.2 pg/mL, normal values of thyroid-stimulating hormone and free thyroxin) is frequently observed in patients with acute myocardial infarction (AMI; generally, between 24 and 36 h after pain onset), associated with myocardial damage and poor prognosis [1,2,3]. Interestingly, a recent study showed that patients with LT3S had higher all-cause mortality, whereas those with other mild thyroid dysfunction forms, such as subclinical hypothyroidism and subclinical hyperthyroidism, did not exhibit significantly increased mortality [4]. A large amount of experimental studies showed the multidimensional TH effect on cardiac function and morphology [5,6]. An altered TH metabolism induced interstitial remodeling associated with a pro-fibrotic effect that reversed with T3 and T4 treatment [7]. Moreover, reverse post-ischemic cardiac remodeling can be associated with the TH-positive properties of pro-angiogenesis. Additionally, in this case, chronic hypothyroidism induces rarefaction of the coronary microvasculature, and, thus, vasodilation impairment, but this effect is reversed with T3 administration that promotes remodeling of coronary resistance vessels [8]. Furthermore, TH metabolic changes induce bioenergetic remodeling of mitochondria, and T3 treatment in experimental AMI settings is associated with the protection of mitochondrial integrity and energy mechanisms [9]. In addition, TH cardioprotection is mediated by regulation of prosurvival pathways. In particular, Pantos C et al. showed a T3 antiapoptotic TH effect mediated by a decrease in p38 MAPK activation [10]. Interestingly, Mouruzis et al. showed a dose-dependent effect of T3 on Akt phosphorylation; in fact, mild activation induced by a lower T3 dosage resulted in favorable effects, whereas further induction of Akt signaling by higher doses of TH was accompanied by increased mortality and ERK activation [11]. Thus, due to the experimental evidence of a cardioprotective role of thyroid hormone (TH) in AMI models, a rising topic of discussion is the necessity to focus on a correct dosage in the therapeutic treatment with TH [2]. Low vitamin D has been associated with cardiovascular risk and adverse outcomes [12]. Moreover, recently, a high incidence of hypovitaminosis D has been documented in AMI patients [13]. Experimental data suggested a beneficial role of vitamin D in the thyroid profile, through the improvement of deiodinase 2 (D2) expression [14]. Vitamin D supplementation is safe and very rarely toxic, even at high doses, and could represent a reliable alternative to TH treatment [15]. Thus, since there are no data on the relationship between hypovitaminosis D and LT3 syndrome in AMI clinical settings, we firstly assessed whether there is a relationship between LT3 and hypovitaminosis D in AMI patients.

## 2. Methods

This was an observational, prospective, non-interventional study. One hundred and twenty-four AMI patients were enrolled at the Ospedale del Cuore G. Pasquinucci-Clinical Cardiology Department (Massa, Italy; latitude coordinate 44° N). Patients were considered eligible to be enrolled in the study on the basis of inclusion criteria that were as follows: (1) male and female patients, of all ethnicities, admitted to the CCU for chest pain and subsequently proven STEMI; (2) adult subjects; (3) patients subject to percutaneous coronary revascularization and stenting of the culprit lesion alone within 24 h from the onset of symptoms. All patients were treated according to the existing guidelines for AMI management [16]. The mean time between symptoms of AMI/admission to the CCU unit and randomization was 12 ± 3 h. Exclusion criteria were the following: (1) previous myocardial infarction; (2) previous evidence of moderate-to-severe compromised left ventricular function (ejection fraction < 40%); (3) severe systemic diseases; (4) systemic inflammatory autoimmune disease; (5) patients refusing or unable to supply written informed consent. Moreover, due to their potential interference with TH metabolism and result interpretation, patients already assuming the following drugs were excluded from the study: (1) TH replacement therapy, anti-thyroid drugs; (2) amiodarone; (3) corticosteroids; (4) oral anticoagulant therapy; (5) sympathomimetic drugs; (6) oral contraceptives or estro-progestinic hormone replacement; (7) potentially hepatotoxic drugs (e.g., metotrexate).

Blood samples were taken at admission, and at 3, 12, 24, 48, and 72 h after admission. After rapid centrifugation of a blood sample from an antecubital vein, free triiodothyronine (fT3) was measured immediately after the blood sample using a completely automated AIA 600 system (Tosho Corp, Tokyo, Japan), our laboratoy reference intervals being 2–4.4 ng/mL. LT3 was defined by a value of fT3 ≤ 2.2 pg/mL, occurring within 3 days of hospital admission [1]. Quantitative determination of 25-hydroxyvitamin D (25(OH)D) was performed by DiaSorin “LIAISON 25-OH Vitamin D TOTAL” CLIA, a direct competitive immunochemiluminescent assay, as we previously described in detail [17,18]. Although the gold standard for 25(OH)D measurement remains mass spectrometry, the LIAISON automated immunoassay is largely utilized in clinical laboratories, simple to use, and reliable, as LIAISON measurements have been proved to be well correlated with LC-MS-MS measurements [19].

Levels were defined as follows: sufficiency as a value of ≥30 ng/mL, vitamin D insufficiency as 25(OH) D between 21 and 29 ng/mL, deficiency in 25(OH)D as below 20 ng/mL, and severe deficiency as values under 10 ng/mL [20].

However, other scientific societies (e.g., Institute of Medicine, OM, USA) utilize other vitamin D cut-offs and define 25(OH)D of ≥ 20 ng/mL as sufficiency; between 19 and 12 mg/mL as insufficiency; and <12 ng/mL as deficiency, as recommended in their “Dietary Reference Intakes” [11]. To note, even though these limits are thought to ensure skeletal health (e.g., minimized PTH levels), the appropriate levels of 25(OH)D which should be maintained to prevent other vitamin D-related extra-skeletal diseases still remain to be determined [21,22].

This study protocol was carried out in accordance with Good Clinical Practice and Declaration of Helsinki statements concerning medical research in humans. The local ethical committee approved this study protocol (study registration number 266).

Continuous variables were summarized as mean ± SD, or percentage for categorical variables. Statistical analyses were performed using the statistical package Statview, version 5.0.1 (SAS Institute, Abacus Concept, Inc., Berkeley, CA, USA) and included Student’s *t* test, χ^2^ test, ANOVA analysis, and Scheffe’s test. Among the variables reported in Table 1, those with a univariate significant association were entered in the multivariate logistic regression analysis to identify independent variables for LT3S. A two-tailed *p*-value < 0.05 was chosen as the level of significance.

## 3. Results

The characteristics of the patient population are reported in Table 1. The mean plasma 25(OH)D in the overall population resulted in being 21 ± 11 ng/mL, well below the generally recommended threshold, corresponding to 30 ng/mL. In the overall population, only 21 patients (17%) had sufficient 25(OH)D levels, whereas 106 (83%) patients revealed a suboptimal 25(OH)D concentration, with 65 (52%) having 25(OH)D deficiency (<20 ng/mL).

As levels of 25(OH)D did not change during the first 72 h of evaluation (from 12 ± 7, 12 ± 7, 12 ± 8, 12 ± 9, 12 ± 9, and 12 ± 5 ng/mL at admission, and 3, 12, 24, 48, and 72 h after admission), values at admission were considered for 25(OH)D.

The percentage of subjects with severe 25(OH)D deficiency was significantly higher in the LT3 group (33% vs. 13%, *p* = 0.007, χ2 test; Figure 1). When LT3S was evaluated as a dependent variable, severe 25(OH)D deficiency (OR 2.6: 95%CI 1–6.7, *p* < 0.05) remained as an independent determinant after logistic multivariate adjustment together with age (>69 yrs, 50th percentile; OR 3.4, 95%CI 1.3–8.3, *p* < 0.01), but not female gender (OR 1.7, 95%CI 0.7–4.2, *p* = ns).

## 4. Discussion

This pilot study shows, for the first time, a relationship between hypovitaminosis D and LT3 in AMI. In particular, 25(OH)D deficiency was associated with a higher incidence of LT3 and was an independent predictor of this syndrome. Accordingly, although the reduction in T4-to-T3 peripheral conversion can be restored after a few days from the onset of an acute clinical event, the presence of 25(OH)D deficiency might further delay this conversion. This is related to the fact that vitamin D counteracts the reduction in the peripheral conversion of T4 into T3 through the increase in D2 expression [14]. This action directly acts on one of the mechanisms inducing LT3 in AMI that consists in increased D3 and reduced D1 and D2 activities [2]. In physiological conditions, D2 is considered the major component responsible for the daily synthesis of T3 in the myocardium; however, Pol et al., in an AMI experimental model, showed that myocardium D3 activity increase is strongly associated with post-AMI left ventricular remodeling [23,24]. More recently, Sabatino et al., in an experimental rat model of ischema/reperfusion, showed that T3 administration induced a major expression of D2 and D1 in the myocardial area at risk, and this was associated with the attempt of the heart to preserve as many functional cardiac parameters as possible [25]. Thus, taking into account data on cardiac deiodinases and the possible association with vitamin D effects, it is reasonable to hypothesize that vitamin D deficiency may impair TH metabolism, and, on the other side, vitamin D supplementation may favor euthyroidism restoration.

The challenge to treat AMI patients with TH replacement therapy is a current focus of clinical research. Actually, there is no consensus to treat AMI patients with thyroid replacement therapy with the goal to restore euthyroidism. However, many aspects are under evaluation in this field, which include the type of patients who truly benefit from TH treatment, patients with or without TH metabolism abnormalities, the type of TH to administer (T3 or T4 or a combination of T3 and T4), and the timing of when to start and interrupt this treatment. However, the main difficulty and object of discussion is the need to adopt and maintain a low T3 regimen dose, in order to reduce the risk of a pharmacological-induced supraphysiological T3 concentration that could lead to increased mortality and other adverse outcomes, as documented in experimental and clinical settings [11,26]. The presence of hypovitaminosis D could also be the only marker of critical clinical condition in AMI settings, as suggested by the evidence that hypovitaminosis D has been associated with a higher incidence of cardiovascular mortality and morbidity [27]. Moreover, a recent study showed that a lower 25(OH)D3 level on admission was associated with the higher no-reflow phenomenon, which is considered a severe and irreversible type of myocardial damage occurring in AMI settings [28]. Thus, the association between hypovitaminosis D and T3 opens potential therapeutic challenges, aiming at the restoration of euthyroidism and normalization of hypovitaminosis D through vitamin D administration. In fact, beyond the vitamin D-related effect on D2, other possible mechanisms related to vitamin D need to be investigated such as its antioxidant activity which can counteract the acute derangements in TH levels produced by the oxidative stress in AMI. That is: killing two birds with one shot. Further studies are needed to reinforce these preliminary data and to ascertain the potential therapeutic challenges. However, the relationship between 25(OH)D and LT3S in AMI has potential clinical relevance, opening the possibility to counteract LT3S, restoring T3 levels, and improving outcomes in acute and long-term periods without the adverse effects related to TH treatment, especially in patients with severe 25(OH)D deficiency.

## Figures and Tables

**Figure 1 jcm-10-05267-f001:**
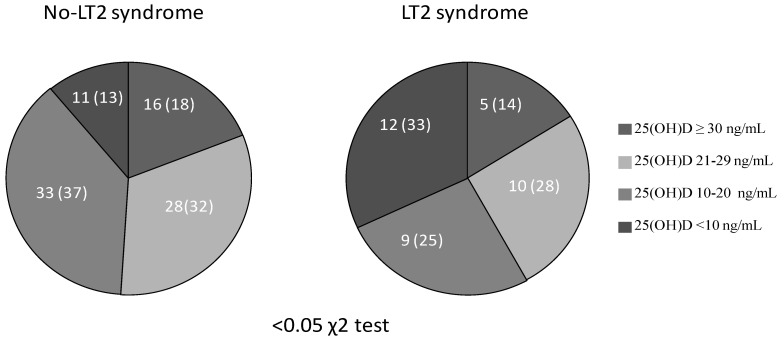
The percentage of subjects with 25(OH)D deficiency in patients with and without low T3 syndrome.

**Table 1 jcm-10-05267-t001:** Demographic, clinical, and laboratory characteristics of the overall population with acute myocardial infarction (AMI).

Demographic, Clinical and Instrumental Parameters	LT3S	No-LT3S	*p* Value
N	36	88	
Age (years)	76 ± 10	66 ± 12	<0.001
Males	20 (56)	67 (76)	<0.05
BMI (kg/m2)	28 ± 6	27 ± 5	Ns
Current smokers	6 (18)	28 (33)	Ns
Type 2 Diabetes	9 (26)	14 (16)	Ns
Hypertension	24 (68)	53 (60)	Ns
Dyslipidemia	17 (49)	45 (51)	Ns
STEMI	27 (82)	76 (88)	Ns
EF (%)	49 ± 10	49 ± 8	Ns
WMSI	1.4 ± 0.2	1.3 ± 0.3	Ns
Ca (mg/dL)	8.5 ± 0.4	8.9 ± 0.4	Ns
K (mEq/L)	4.1 ± 0.5	4.1 ± 0.4	Ns
Mg (mEq/L)	1.6 ± 0.2	1.6 ± 0.2	Ns

Data are reported as mean (SD) or n (%). LT3S, Low T3 Syndrome; BMI, body mass index; STEMI, ST-elevation myocardial infarction; EF, ejection fraction; WMSI, wall motion score index; Ca, calcium; K, potassium Mg, magnesium.

## Data Availability

The data of this study are available to pingi@ifc.cnr.it, cristina.vassale@ftgm.it.

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
