# Peer review of "Hypovitaminosis D and Low T3 Syndrome: A Link for Therapeutic Challenges in Patients with Acute Myocardial Infarction"

_jcm, 2021, doi:10.3390/jcm10225267_

Round 1

Reviewer 1 Report

The paper sounds novel and interesting. Some minor points need to be addressed in the discussion:

1) It is needed to clarify that 30 ng/ml is not a universally accepted cut-off for hypovitaminosis D.  Many societies consider 20 ng/ml the real cut-off. It is acceptable indeed, that for extra skeletal effects the level needs to be higher.

2)This causal effect sounds very interesting. However, it is necessary to admit in the discussion that a "reversal causality" can happen with vitamin D and hypovitaminosis D could be a only marker of poor condition in chronic/acute patients... until causality is proven.

3) The gold standard for 25OHD measurement is mass spectrometry, even if LIAISON measurements has been proved to be well correlated with LC-MS-MS measure. 

Author Response

1) It is needed to clarify that 30 ng/ml is not a universally accepted cut-off for hypovitaminosis D.  Many societies consider 20 ng/ml the real cut-off. It is acceptable indeed, that for extra skeletal effects the level needs to be higher.

We better defined this point as follow:

Levels were defined as follows: sufficiency for value ±>= 30 ng/mL, vitamin D in-sufficiency as a 25(OH) D between 21–-29 ng/mL, deficiency for 25(OH)D below 20 ng/mL with severe deficiency for values under 10 ng/mL. However, other Scientific Societies (e.g. Institute of Medicine, -OM, USA) utilized other Vitamin D cut-offs, and defined 25(OH)D ± >=20 ng/mL sufficiency; between 19 and 12 mg/mL insufficiency; < 12 ng/mL deficiency, as recommended in their “Dietary Reference Intakes” (11). To note, whether these limits are thought to ensure skeletal health (e.g. minimized PTH levels), the appropriate levels of 25(OH)D, which should be maintained to prevent other vitamin D-related extra-skeletal diseases, remains still to be determined. 

2)This causal effect sounds very interesting. However, it is necessary to admit in the discussion that a "reversal causality" can happen with vitamin D and hypovitaminosis D could be a only marker of poor condition in chronic/acute patients... until causality is proven.

Accordingly, we added the following sentence in the discussion:

The presence of hypovitaminosis D could be a only marker of critical clinical condition, also in AMI setting, as suggested by the evidence that hypovitaminosis D has been associated to higher incidence of cardiovascular mortality and morbidity. Moreover, a recent study showed that lower 25(OH)D3 level on admission was associated with higher no-reflow phenomenon, which is considered a severe and irreversible myocardial damage occurring in AMI setting.  

3) The gold standard for 25OHD measurement is mass spectrometry, even if LIAISON measurements has been proved to be well correlated with LC-MS-MS measure. 

Accordingly, we added the following sentences in the method section:

Quantitative determination of 25(OH)D was performed by DiaSorin “LIAISON 25-OH Vitamin D TOTAL” CLIA, a direct competitive immunochemiluminescent assay, as we previously described in details (9, 10). Although the gold standard for 25(OH)D measurement remains mass spectrometry, LIAISON automated immunoassay is very diffuse in clinical laboratories, more simple to use, and reliable, as LIAISON measurements has been proved to be well correlated with LC-MS-MS measure.

Reviewer 2 Report

This is a clinical study that showed for the first time a relationship between hypovitaminosis D and low T3 syndrome in patiets with myocardial infarction. The study included 123 patients with STEMI. The issue is important with potential therapeutic implications for the treatment of myocardial infarction.

Comments

  1.  In introduction, please discuss in more detailthe effects of T3 on cardiac remodeling and heart failure after AMI
  2. Please add in methods more detailed information about the measurements of thyroid hormones and vitamin D
  3. Please add a table in results comparing demographic, clinical and laboratory charactiristics in the 2 groups: with and without low T3
  4. It would be very interesting to present data of FT4 and FT3 levels at various time points (admission, 3, 12,24, 48, 72 h) in the form of curves between  patients with vitD deficiency and no deficiency
  5. In discussion, please refer to the low T3 syndrome as a syndrome that also affects T3 at the tissue level. In this regard, restoration of tissue T3 might be difficult with T3 administration, but may be achieved with vitD supplementation

Minor comments

1.Please change "withdrawals" wtih "blood samples"

Author Response

  1. In introduction, please discuss in more detail the effects of T3 on cardiac remodeling and heart failure after AMI

A large amount of experimental studies showed the multidimensional TH effect on cardiac function and morphology. An altered TH metabolism induced interstitial remodeling associated with a pro-fibrotic effect that reversed with T3 and T4 treatment. Moreover, a reverse post-ischemic cardiac remodeling can be also associated to the TH positive properties on pro-angiogenesis. In this case, chronic hypothyroidism induces rarefaction of coronary microvasculature, and, thus, vasodilation impairment, but this effect is reversed with T3 administration that promotes remodeling of coronary resistance vessels.  Furthermore, TH metabolic changes induce bioenergetic remodeling of mitochondria, and T3 treatment in experimental AMI setting is associated to the protection of mitochondrial integrity and energy mechanisms. In addition, TH cardioprotection is mediated by regulation of prosurvival pathways. In particular, Pantos C et al showed a T3 antiapoptotic TH effect mediated by a decrease in p38 MAPK activation. Interestingly, Mouruzis et al. showed a dose-dependent effect of T3 on Akt phosphorylation, in fact, mild activation induced by lower T3 dosage resulted in favorable ef-fects, whereas further induction of Akt signaling by higher doses of TH was accompanied by increased mortality and ERK activation.

  1. Please add in methods more detailed information about the measurements of thyroid hormones and vitamin D

After rapid centrifugation of a blood sample from an antecubital vein, free triiodothyronine (fT3), was measured immediately after the blood sample using a completely automated AIA 600 system (Tosho Corp, Tokyo, Japan), our laboratoy reference intervals being 2-4.4 ng/mL. LT3 was defined by a value of fT3 ≤ 2.2 pg/mL, occurring within 3 days of hospital admission (1). Quantitative determination of 25(OH)D was performed by DiaSorin “LIAISON 25-OH Vitamin D TOTAL” CLIA, a direct competitive immunochemiluminescent assay, as we previously described in details (9, 10).

  1. Please add a table in results comparing demographic, clinical and laboratory charactiristics in the 2 groups: with and without low T3

As suggested, we added demographic, clinical and laboratory characteristics of the 2 groups according to low T3 in the table 1

  1. It would be very interesting to present data of FT4 and FT3 levels at various time points (admission, 3, 12,24, 48, 72 h) in the form of curves between  patients with vitD deficiency and no deficiency

This is a very interesting issue, but it was not the focus of our study. However, when we used the cut-off of 30 ng/mL we did not observe any difference in the curves between pts with and without vitamin D deficiency. Indeed, patients with severe vitamin D deficiency are particularly interesting for further assessment of this aspect. Thus, a specific study should be designed for this purpose, including a larger number of patients with severe Vit D deficiency.

  1. In discussion, please refer to the low T3 syndrome as a syndrome that also affects T3 at the tissue level. In this regard, restoration of tissue T3 might be difficult with T3 administration, but may be achieved with vitD supplementation

This is an important point and we answered as follows:

Accordingly, although the reduction of T4 to T3 peripheral conversion, can start up again after few days from the onset of the acute clinical event, the presence of 25(OH)D deficiency, may further delay this conversion. This is related to the fact that vitamin D counteracts the reduction of the peripheral conversion of T4 into T3 through the increase in D2 expression (7). This action directly acts on one of the mechanisms inducing LT3 in AMI that consists in increased D3 and reduced D1 and D2 activities (2). In physiological conditions, D2 is considered the major responsible of daily synthe-sis of T3 in myocardium, however, Pol et al., in an AMI experimental model, showed that myocardium D3 activity increases is strongly associated with post-AMI left ventricular remodeling. More recently, Sabatino et al., in an experimental rat model of ischema/reperfusion, showed that T3 administration induced a major expression of D2 and D1 in the myocardial area at risk, and this was associated to the attempt of the heart to preserve as much as possible, functional cardiac parameters. Thus, taking into account data on cardiac deiodinases and possible association with vitamin D effects, it is reasonable to hypothesize that vitamin D deficiency may impair TH metabolism and, on the other side, vitamin D supplementation may favour euthyroidism restoration.

Minor comments

1.Please change "withdrawals" with "blood samples"

Accordingly, we made changes as suggested by the reviewer

Reviewer 3 Report

Hypovitaminosis D and low T3 syndrome.

Pingitore et al

This is a short paper describing the possible relationship between low Vitamin D status and impaired conversion from T4 to T3 in acutely unwell MI patients. The authors have demonstrated a direct link between these two variables and that addresses the major concern we all have with such papers – proving a “cause and effect” relationship as opposed to an associative one. It is concise and clearly written and the references are up to date and relevant.

The relationship between acute illness and low T3 is well established and it is generally accepted that the peripheral conversion of T4 to T3 corrects itself when the patient is less acutely unwell. The link between low T3 and low Vitamin D though has not been fully explored and this paper starts that process off.

There are a couple of areas that I would have like to have seen explored a bit more fully:

Were there any patients in the cohort with a pre-existing thyroid disease of any kind i.e. if the ability to convert T4 into T3 was already impaired in some patients, a co-existing low Vitamin D may not be relevant or indeed might even make things worse for such a patient! Could the authors correct for that possible variable?

During the recovery phase from an acute illness, T4 to T3 conversion starts up again. Given the data in this paper, could the authors speculate on whether TFTs would normalise more slowly if the Vitamin D remains low i.e. would it be better to replace the Vitamin D in addition to conventional TH replacement therapy?

Overall though, I enjoyed reading this paper and I actually learned something from it. I would recommend it for publication with a few minor revisions.

Author Response

This is a short paper describing the possible relationship between low Vitamin D status and impaired conversion from T4 to T3 in acutely unwell MI patients. The authors have demonstrated a direct link between these two variables and that addresses the major concern we all have with such papers – proving a “cause and effect” relationship as opposed to an associative one. It is concise and clearly written and the references are up to date and relevant.

The relationship between acute illness and low T3 is well established and it is generally accepted that the peripheral conversion of T4 to T3 corrects itself when the patient is less acutely unwell. The link between low T3 and low Vitamin D though has not been fully explored and this paper starts that process off.

Many thanks for the appreciation of our work.

There are a couple of areas that I would have like to have seen explored a bit more fully:

Were there any patients in the cohort with a pre-existing thyroid disease of any kind i.e. if the ability to convert T4 into T3 was already impaired in some patients, a co-existing low Vitamin D may not be relevant or indeed might even make things worse for such a patient! Could the authors correct for that possible variable?

Many thanks for this question that provided us to add, in the method section, other exclusion criteria we did not include in the first version of the manuscript. Thus, we added the following sentences:

Moreover, due to their potential interference with TH metabolism and results interpretation, patients already assuming the following drugs were excluded from the study: 1) TH replacement therapy, anti-thyroid drugs; 2) amiodarone; 3) corticosteroids; 4) oral anticoagulant therapy; 5) sympathomimetic drugs; 6) oral contraceptives or estro-progestinic hormone replacement; 7) potentially hepatotoxic drugs (e.g. metotrexate).

During the recovery phase from an acute illness, T4 to T3 conversion starts up again. Given the data in this paper, could the authors speculate on whether TFTs would normalise more slowly if the Vitamin D remains low i.e. would it be better to replace the Vitamin D in addition to conventional TH replacement therapy?

As suggested, we added the following sentences:

Accordingly, although the reduction of T4 to T3 peripheral conversion, can start up again after few days from the onset of the acute clinical event, the presence of 25(OH)D deficiency, may further delay this conversion. This is related to the fact that vitamin D counteracts the reduction of the peripheral conversion of T4 into T3 through the increase in D2 expression (7).

Overall though, I enjoyed reading this paper and I actually learned something from it. I would recommend it for publication with a few minor revisions.

Again, many thanks